# Efficient Modeling of Long-range fMRI Dynamics with a 2D Natural Image Autoencoder

## Abstract

Modeling long-range spatiotemporal dynamics in functional Magnetic Resonance Imaging (fMRI) remains a key challenge due to the high dimensionality of the four-dimensional signals. Prior voxel-based models, although demonstrating excellent performance and interpretation capabilities, are constrained by prohibitive memory demands and thus can only capture limited temporal windows. To address this, we propose TABLeT (Two-dimensionally Autoencoded Brain Latent Transformer), a novel approach that tokenizes fMRI volumes using a pre-trained 2D natural image autoencoder. Each 3D fMRI volume is compressed into a compact set of continuous tokens, enabling efficient long-sequence modeling with a simple Transformer encoder. Across large-scale benchmarks including the UK-Biobank (UKB), Human Connectome Project (HCP), and ADHD-200 datasets, TABLeT outperforms existing models in multiple tasks, while demonstrating substantial gains in computational and memory efficiency over the state-of-the-art voxel-based method. Furthermore, we demonstrate that TABLeT can be pre-trained with a self-supervised masked token modeling approach, improving downstream tasks' performance. Our findings suggest a promising approach for scalable spatiotemporal modeling of brain activity.

## 1 Introduction

The human brain is a spatiotemporal dynamic system whose activity can be non-invasively measured using functional magnetic resonance imaging (fMRI). A large body of work has leveraged fMRI to investigate functional connectivity patterns for tasks such as neurological disorder diagnosis or demographic attribute prediction (Kawahara et al., 2017; Kan et al., 2022; Popov et al., 2024; Malkiel et al., 2022; Kim et al., 2023; Caro et al., 2024; Dong et al., 2024). Existing approaches can be broadly divided into two categories: *ROI-based methods* and *voxel-based methods*.

ROI-based methods first define a set of regions of interest (ROIs) based on anatomical segmentation (Power et al., 2011), extract their corresponding time-series signals, and then compute functional connectivity (FC) matrices as model inputs. Although this approach is computationally efficient for managing the high dimensionality of fMRI data, it has several limitations: performance strongly depends on the choice of ROIs, fine-grained 3D spatial structures may be lost, and aggressive compression can discard informative signals. To overcome these limitations, voxel-based methods such as TFF (Malkiel et al., 2022) and SwiFT (Kim et al., 2023) have been proposed. These methods directly process raw 4D fMRI data, thereby preserving spatial and temporal information, while also allowing detailed interpretation as they directly operate on the given image. However, due to the massive scale of fMRI volumes, the temporal length that could be simultaneously processed by the model is severely restricted (e.g., TFF and SwiFT use only 20 timesteps at once), potentially missing informative long-range temporal dynamics, and limiting use for tasks that require longer-range interactions, such as the infraslow BOLD–LFP coupling and global arousal waves that unfold over tens of seconds (Pan et al., 2013; Raut et al., 2021).

In this work, we aim to improve voxel-based fMRI modeling by *tokenizing* fMRI volumes into a compact set of continuous tokens, thereby enabling Transformers (Vaswani et al., 2017) to model substantially longer temporal sequences. To this end, we paid attention to the remarkable perceptual information preservation capability of the Deep Compression Autoencoder (DCAE) (Chen et al., 2025) and aimed to leverage it, as it effectively tokenizes a $256 \times 256$ 2D natural image into *just* 64

continuous tokens (a compression ratio of 32). Motivated by this, we ask *whether a high-performing 2D autoencoder trained on **natural images** can serve as an effective tokenizer for **4D fMRI** data.*

Our findings reveal that such an autoencoder can indeed be applied to tokenize fMRI volumes. By rearranging the tokens extracted from each 2D slice of a 3D fMRI volume, we compress an entire volume into *only* 27 continuous tokens, thereby dramatically reducing the input size and enabling efficient long-sequence modeling with a simple Transformer encoder-based architecture. We dub our method TABLeT, **T**wo-dimensionally **A**utoencoded **B**rain **L**at**e**nt **T**ransformer, which achieves superior performance compared to both ROI-based and voxel-based baselines on demographic attribute prediction and attention-deficit hyperactivity disorder (ADHD) diagnosis tasks, while drastically saving memory and computation costs compared to the voxel-based baseline. Moreover, we show that TABLeT benefits from a self-supervised masked token modeling approach that pre-trains the model on unlabeled fMRI data, further boosting downstream task performance beyond models trained from scratch.

## 2 RELATED WORK

**ROI-Based Methods.** ROI-based methods parcellate the brain into ROIs and average the BOLD signals within each. The signals are transformed into FC matrices by computing the correlation between the time series of the ROIs. BrainNetCNN (Kawahara et al., 2017) treats the FC matrix as a 2D image and uses edge-to-edge, edge-to-node, and node-to-graph convolutional filters to utilize topological locality in ROI-based networks. Brain Network Transformer (Kan et al., 2022) adapts the Transformer architecture to process FC matrices as graphs of ROIs. meanMLP (Popov et al., 2024) is a lightweight MLP-based model that applies the same MLP repeatedly across parcellated fMRI time-series and averages the resulting embeddings across time before a final classification layer. Brain-JEPA (Dong et al., 2024) is a joint-embedding predictive architecture (JEPA) model pretrained on parcellated fMRI with spatiotemporal masking and gradient-based positioning. Even though computationally efficient, they are inherently limited by the strong pre-processing step that turns brain signals into FC matrices; it is heavily influenced by the choice of ROIs, and during the process, structural information as well as other signals can be discarded.

**Voxel-Based Methods.** Voxel-based methods process 4D fMRI volumes, enabling end-to-end learning of spatiotemporal features without ROI aggregation. TFF (Malkiel et al., 2022) operates on entire 4D volumes using a two-phase approach: self-supervised pretraining to reconstruct 3D volumes and fine-tuning. It captures fine-grained spatiotemporal dynamics, enabling transfer learning from unlabeled data. SwiFT (Kim et al., 2023) extends the Swin Transformer to 4D fMRI volumes with a 4D window multi-head self-attention mechanism and absolute positional embeddings. Voxel-based methods are free from the issues with ROI-based models; however, they are burdened with a higher memory and computation load, as they need to deal with high-dimensional data.

**Self-Supervised Pretraining.** Self-supervised learning (SSL) has emerged as a powerful pre-training framework for vision models, enabling scalable representation learning from unlabeled datasets. MAE (He et al., 2022) introduces an asymmetric encoder-decoder architecture, where a high portion of input image patches are masked. Visible patches are encoded by Vision Transformer (ViT), and the decoder reconstructs the masked patches in pixel space. MAE demonstrated superior downstream task transfer, such as classification and segmentation. SimMIM (Xie et al., 2022) proposes a masked image modeling (MIM) framework using hierarchical transformers and a simple linear prediction head. VideoMAE (Tong et al., 2022) extends the MAE framework to videos by randomly applying masks on spatio-temporal cubes across spatio-temporal dimensions. The model reconstructs the masked cubes, learning dynamics, and long-range interactions.

**Deep Compression Autoencoder.** Deep Compression Autoencoder (DCAE) (Chen et al., 2025) introduces an autoencoder framework for accelerating high-resolution diffusion models through extreme spatial compression ratios of up to $128\times$. DCAE achieves superior reconstruction quality at high compression levels by residual autoencoding. Residual autoencoding utilizes non-parametric shortcuts that enable the model to learn residuals. The encoder downsample blocks adapt a space-to-channel operation, and the decoder upsample blocks use a channel-to-space operation. These non-parametric operations effectively preserve information without learned parameters.

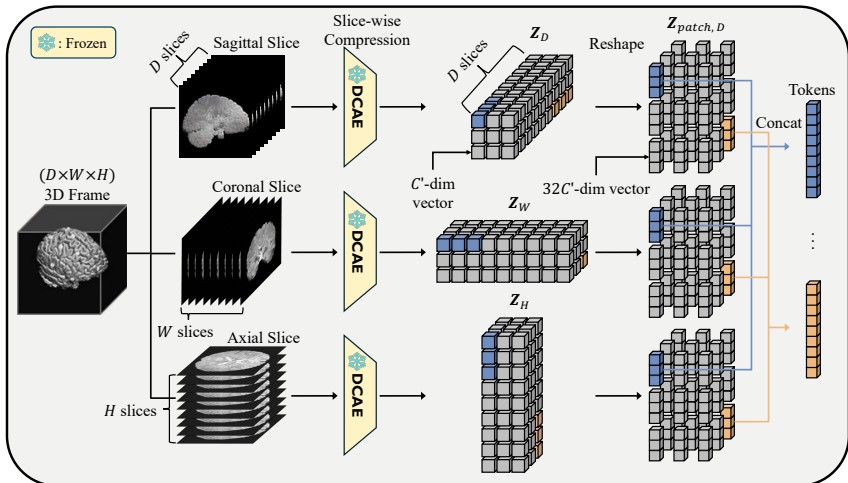

(a) Tokenization process of a 3D volume using a 2D encoder.

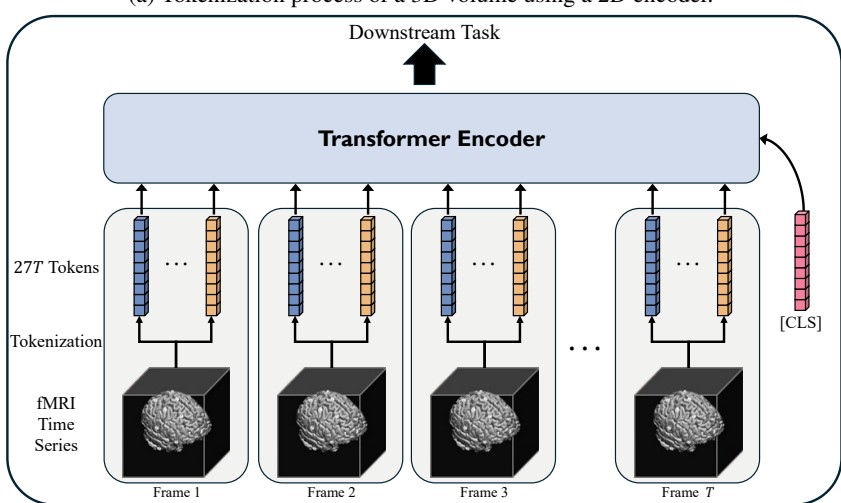

(b) Overview of TABLeT.

Figure 1: In TABLeT, each frame of the fMRI timeseries is tokenized by a 2D autoencoder, and the resulting tokens are processed by a Transformer.

## 3 METHOD

### 3.1 TOKENIZATION OF FMRI WITH 2D NATURAL IMAGE AUTOENCODER

To develop a more efficient approach for voxel-based fMRI modeling, our goal was to design a tokenizer that could substantially compress fMRI voxels while minimizing information loss. We chose to employ the encoder part of an autoencoder, since it can handle tokenization while preserving coarse spatial topology. A straightforward strategy would be training an autoencoder directly on fMRI data. However, this approach is both computationally prohibitive and data hungry because reliable training may require large sample sizes that are rarely available in medical imaging. Moreover, the resulting models often generalize poorly, as fMRI characteristics vary across scanners and acquisition protocols.

To circumvent these challenges, we sought a training-free tokenization scheme that also preserves the fidelity of the original signal. Inspired by the recent advances in image autoencoders, we hypothesized that such models could also serve as effective tokenizers for fMRI volumes. Among existing options, we adopt DCAE, which achieves strong compression while maintaining image details. Specifically, we employ the unmodified *dc-ae-f32c32-in-1.0* checkpoint provided by Chen et al. (2025) for all 2D natural image DCAE experiments.

We first compared the reconstruction performance, on fMRI brain data, of an off-the-shelf 2D natural image DCAE (hereafter 2D DCAE) with a 3D DCAE trained directly on fMRI data (hereafter 3D DCAE), as detailed in Sec. 4.4. One important thing to note is that fMRI data are timeseries of 3D images, while the 2D DCAE only operates with 2D images. Therefore, we slice the data into 2D images and independently feed them into the autoencoder. Surprisingly, the pre-trained 2D DCAE produced higher-quality reconstructions despite never being trained on fMRI. Based on this finding, we propose to tokenize each 3D volume independently using the 2D DCAE encoder and apply this procedure across the entire fMRI sequence, as described below.

**Tokenization of a 3D Volume with Slicing.**  Each fMRI frame is a 3D volume $\mathbf{X} \in \mathbb{R}^{1 \times D \times H \times W}$. The single channel is first duplicated across three channels to simulate an RGB structure, giving $\mathbf{X} \in \mathbb{R}^{3 \times D \times H \times W}$. One spatial dimension is then chosen as the slicing axis, so the volume becomes a stack of 2D images. For example, if we slice by the depth axis, the volume is treated as $D$ images of shape $\mathbb{R}^{3 \times H \times W}$. Each image slice is compressed independently into a latent representation $\mathbf{Z} \in \mathbb{R}^{C' \times \frac{H}{32} \times \frac{W}{32}}$, where the factor of 32 is the DCAE's spatial compression ratio.

**Aggregation of 3 Axes.**  This procedure is repeated for all three slicing axes, producing three latent volumes: $\mathbf{Z}_D \in \mathbb{R}^{D \times C' \times \frac{H}{32} \times \frac{W}{32}}$,   $\mathbf{Z}_H \in \mathbb{R}^{H \times C' \times \frac{D}{32} \times \frac{W}{32}}$,   $\mathbf{Z}_W \in \mathbb{R}^{W \times C' \times \frac{D}{32} \times \frac{H}{32}}$. As we want to align the three possible latent volumes, the latents are grouped and concatenated along the uncompressed dimension (the slicing axis) by patches of size 32, reshaping them to $\mathbf{Z}_{\text{patch},D}, \mathbf{Z}_{\text{patch},H}, \mathbf{Z}_{\text{patch},W} \in \mathbb{R}^{32C' \times \frac{D}{32} \times \frac{H}{32} \times \frac{W}{32}}$. This yields $\frac{D}{32} \times \frac{H}{32} \times \frac{W}{32}$ tokens for each slicing variation, where each token corresponds to a position in the downsampled 3D grid and has a hidden dimension $32C'$. The tokens from the three variations are then concatenated with the tokens from other variations that belong to the same spatial position, resulting in $\frac{D}{32} \times \frac{H}{32} \times \frac{W}{32}$ tokens per frame with hidden dimension $96C'$. In our case, $H = W = D = 96$ and $C' = 32$, which means each 3D volume of shape $(1, 96, 96, 96)$ is tokenized into 27 tokens with an embedding dimension of 3072. Finally, we note that tokenization is performed only once, and the tokens are cached for later use, making its computational cost negligible compared to the subsequent training process.

## 3.2 TABLeT Model Architecture

To capture the spatiotemporal dynamics of tokenized fMRI sequences, we design a simple yet effective Transformer encoder (Vaswani et al., 2017), naming the pipeline **TABLeT** (Two-dimensionally Autoencoded Brain Latent Transformer). The architecture is built on a standard Transformer encoder backbone and integrates several modern components commonly adopted in large language models (Qwen et al., 2025; Grattafiori et al., 2024). In particular, we adopt grouped query attention (Ainslie et al., 2023) to efficiently handle long sequences, along with the rotary positional encoding (Su et al., 2024). In addition, we employ `F.scaled_dot_product_attention` from PyTorch (Paszke et al., 2019), which offers both speed and memory savings. Before being fed into the Transformer, fMRI tokens are normalized and projected into a lower-dimensional embedding space via a linear layer. A `[CLS]` token is prepended to the sequence, followed by an additional normalization step to enhance training stability. Unless stated otherwise, the model is composed of 12 Transformer layers with 14 attention heads and 2 key–value heads, processing sequences of tokens from 256 volumes at once ($T = 256$). We randomly sampled 256 frames from the entire sequence of each subject at every training iteration, while for validation, we used all of the frames by partitioning the sequence and averaging the outputs across partitions, following Kim et al. (2023).

## 3.3 Self-supervised Pre-training with Masked Token Modeling

Taking inspiration from SimMIM (Xie et al., 2022), we leverage a *masked token modeling* approach to pre-train the Transformer encoder of TABLeT. The idea is to encourage the model to learn meaningful spatiotemporal representations from partially observed fMRI sequences. Starting from the tokens created by the 2D DCAE, we randomly mask some of the tokens by replacing them with a `[MASK]` token. From the partially masked input tokens, we task the Transformer encoder to predict the masked tokens by passing the output tokens through a linear prediction head that reconstructs the input tokens. The model is trained through an $\mathcal{L}_1$ loss exclusively on the masked tokens:

$$L = \frac{1}{\Omega(\mathbf{Z}_M)} ||\mathbf{y}_M - \mathbf{Z}_M||_1 \tag{1}$$

Where $\mathbf{Z}, \mathbf{y} \in \mathbb{R}^{96C' \times \frac{D}{32} \times \frac{H}{32} \times \frac{W}{32}}$ are the input tokens and the predicted tokens, respectively. The subscript $M$ denotes the set of masked tokens, and $\Omega$ counts the number of elements (thus the number of masked tokens). We used a masking ratio of 0.5 as the default in our experiments.

Even though it is possible to use a typical masked *image* modeling approach by masking the brain volume directly, we chose masked token modeling as it is much more computationally efficient, and it still performs well in practice, as we do not change the DCAE encoder during fine-tuning.

**Masking Strategy.** We mask the input tokens with a learnable mask token, following masked modeling approaches such as BERT (Devlin et al., 2019), BEiT (Bao et al., 2022), and SimMIM (Xie et al., 2022). Also, instead of masking the tokens in a completely random manner, the same masking pattern from a single frame is repeated across different frames, similar to the tube masking strategy found within VideoMAE (Tong et al., 2022). This is a measure to prevent the model from "cheating" by looking at tokens in the same location from different frames during reconstruction.

## 4 EXPERIMENTAL RESULTS

### 4.1 EXPERIMENTAL SETTING

**Datasets.** We used resting-state fMRI data from 8,178 middle-aged and older adults from UK-Biobank (UKB) (Sudlow et al., 2015), from 1,061 healthy young adults in the Human Connectome Project (HCP) (Smith et al., 2013), and from 533 children and adolescents, including both individuals diagnosed with ADHD and healthy controls, included in ADHD-200 (Bellec et al., 2017).

For UKB and HCP, we used the preprocessed data provided by UK-Biobank (Miller et al., 2016; Alfaro-Almagro et al., 2018) and HCP (Smith et al., 2013), which goes through the preprocessing pipeline including bias field reduction, skull-stripping, cross-modality registration, and spatial normalization to the MNI space (Evans et al., 1993). For ADHD-200, we used the fMRIPrep (Esteban et al., 2019; 2020) processed data from Bellec et al. (2017) and regressed out nuisance variables using cosine bases, six motion parameters, and aCompCor components. Following Kim et al. (2023), we set each fMRI volume to the shape of $(96, 96, 96)$ by cropping out the background and padding appropriately, and we apply global z-normalization following Malkiel et al. (2022).

We split UKB and HCP using stratified sampling: by age and sex for UKB, and by age, sex, and intelligence score for HCP. For the ADHD-200 dataset, we performed stratified sampling based on diagnosis labels and image acquisition sites, following Kan et al. (2022). We generated four different random stratified splits, and for all of the splits, the training, validation, and test sets were assigned in a 0.7:0.15:0.15 ratio. For the ADHD-200 dataset, we experimented with three random training seeds for each split to ensure reliable results, given the relatively small size of the dataset.

**Prediction Targets and Evaluation Metrics.** We considered sex and age for both UKB and HCP, intelligence (`CogTotalComp-AgeAdj`) for HCP, and diagnosis for ADHD-200. The continuous targets (age, intelligence) are z-normalized using with the training set. Classification tasks were evaluated with accuracy, AUC (Area Under ROC Curve), and F1 score. Regression tasks were evaluated with MAE (Mean Absolute Error), MSE (Mean Squared Error), and $\rho$ (Pearson's correlation).

**Baselines.** We considered five ROI-based models as our baseline: XGBoost (eXtreme Gradient Boosting) (Chen & Guestrin, 2016), BrainNetCNN (Kawahara et al., 2017), Brain Network Transformer (BNT) (Kan et al., 2022), meanMLP (Popov et al., 2024), and Brain-JEPA (Dong et al., 2024). For the Brain-JEPA, we considered a model trained from scratch for a fair comparison. To preprocess the data, we first construct the FC matrix using a total of 450 ROIs, comprising 400 ROIs from the Schaefer-400 atlas (Schaefer et al., 2018) and 50 additional ROIs from the Tian-Scale III atlas (Tian et al., 2020). For the XGBoost model, we used the upper-triangular part of the FC matrix as the input. We followed the preprocessing pipeline of Brain-JEPA for its experiments.

For the voxel-based baselines, we adopted TFF (Malkiel et al., 2022) and SwiFT (Kim et al., 2023), the state-of-the-art voxel-based model. We reproduced the original model with 20 input time frames ($T = 20$) for both of them. We also extended SwiFT to our hardware limit ($T = 50$) to observe possible gains from a longer temporal context; alongside the number of input time frames, the temporal window size was also extended from 4 to 10.

## 4.2 MAIN RESULTS

Table 1: Performance comparison to baselines on classification and regression tasks. The best results are **bolded** and the second best results are underlined.

| Method | UKB | | | | | | ADHD-200 | | |
|---|---|---|---|---|---|---|---|---|---|
| | Sex | | | Age | | | Diagnosis | | |
| | ACC | AUC | F1 | MSE | MAE | $\rho$ | ACC | AUC | F1 |
| XGBoost | 84.1 | 0.916 | 0.830 | 0.698 | 0.686 | 0.553 | 62.3 | 0.650 | 0.555 |
| BrainNetCNN | 91.7 | 0.969 | 0.912 | 0.597 | 0.618 | 0.647 | 59.2 | 0.640 | 0.545 |
| BNT | 92.4 | 0.980 | 0.919 | 0.540 | 0.588 | 0.685 | 63.6 | 0.677 | 0.624 |
| meanMLP | 87.7 | 0.949 | 0.919 | 0.672 | 0.662 | 0.586 | 56.8 | 0.617 | 0.532 |
| Brain-JEPA[1] | 86.8 | 0.943 | 0.862 | 0.688 | 0.669 | 0.574 | – | – | – |
| TFF ($T = 20$) | **98.3** | 0.998 | **0.982** | 0.440 | 0.525 | 0.760 | 63.3 | 0.700 | 0.608 |
| SwiFT ($T = 20$) | 97.4 | 0.998 | 0.972 | 0.366 | 0.480 | 0.800 | 63.3 | 0.603 | 0.623 |
| SwiFT ($T = 50$) | 98.1 | **0.999** | 0.980 | 0.364 | 0.477 | 0.802 | 63.9 | 0.701 | 0.627 |
| TABLeT ($T = 256$) | 97.7 | 0.998 | 0.976 | **0.340** | **0.466** | **0.814** | **65.8** | **0.729** | **0.630** |

| Method | HCP | | | | | | | | |
|---|---|---|---|---|---|---|---|---|---|
| | Sex | | | Age | | | Intelligence | | |
| | ACC | AUC | F1 | MSE | MAE | $\rho$ | MSE | MAE | $\rho$ |
| XGBoost | 82.2 | 0.890 | 0.837 | 0.859 | 0.769 | 0.296 | 0.908 | 0.779 | 0.292 |
| BrainNetCNN | 86.3 | 0.937 | 0.866 | 0.847 | 0.749 | 0.372 | 0.967 | 0.788 | 0.286 |
| BNT | 86.3 | 0.935 | 0.872 | 0.794 | 0.719 | 0.444 | 0.920 | 0.778 | 0.318 |
| meanMLP | 84.5 | 0.915 | 0.855 | 0.846 | 0.751 | 0.370 | 0.887 | 0.767 | 0.340 |
| Brain-JEPA | 73.9 | 0.809 | 0.761 | 0.814 | 0.746 | 0.369 | 0.959 | 0.799 | 0.171 |
| TFF ($T = 20$) | 88.1 | 0.937 | 0.892 | 0.888 | 0.779 | 0.246 | 0.898 | 0.767 | 0.312 |
| SwiFT ($T = 20$) | 93.1 | 0.978 | 0.937 | 0.776 | 0.719 | 0.450 | 0.940 | 0.782 | 0.297 |
| SwiFT ($T = 50$) | 92.2 | 0.972 | 0.929 | **0.764** | **0.699** | 0.460 | 0.865 | 0.758 | 0.354 |
| TABLeT ($T = 256$) | **93.8** | **0.987** | **0.943** | 0.773 | 0.705 | **0.473** | **0.835** | **0.741** | **0.392** |

Tab. 1 presents experimental results comparing the performance of different models on a training-from-scratch setting, and the second-order statistics are detailed in Sec. C. The results demonstrate that TABLeT outperforms baseline methods, including both ROI-based and voxel-based approaches, across four tasks and three datasets, with only marginal gains on the HCP-Age task and competitive performance against voxel-based baselines on the UKB-Sex task.

Interestingly, the results of SwiFT ($T = 20, 50$) and TABLeT indicate a positive association between temporal window length and performance in intelligence prediction and ADHD diagnosis, suggesting that modeling longer temporal variability may be particularly advantageous for these tasks. Sec. 4.6 expands on this observation with a more detailed study.

## 4.3 EFFECT OF PRE-TRAINING ON DOWNSTREAM TASKS

Tab. 2 shows the effectiveness of the masked token pre-training strategy described in Sec. 3.3. We first pre-trained TABLeT on a large UKB dataset with a 9:1 training and validation split. We then fine-tuned the model on HCP and ADHD-200 to simulate a transfer learning setting. For fine-tuning, we only used 10 epochs for HCP and 5 epochs for ADHD-200, which is considerably lower compared to training from scratch.

The results demonstrate that the pre-training of TABLeT indeed contributes to the improvement of downstream task performance, albeit with varying amounts of success depending on the dataset.

---

[1]Since the ADHD-200 dataset contains fMRI data with varying repetition time (TR) values and fewer than 160 frames, the default frame number used in Brain-JEPA, we were unable to conduct experiments.

Table 2: Performance comparison between TABLeT trained from scratch (TFS) and fine-tuned (FT) on HCP and ADHD-200.

| Model | HCP | | | | | | | | | ADHD-200 | | |
|---|---|---|---|---|---|---|---|---|---|---|---|---|
| | Sex | | | Age | | | Intelligence | | | Diagnosis | | |
| | ACC | AUC | F1 | MSE | MAE | $\rho$ | MSE | MAE | $\rho$ | ACC | AUC | F1 |
| TABLeT TFS | 93.8 | **0.987** | 0.943 | 0.773 | 0.705 | 0.473 | 0.835 | 0.741 | 0.392 | **65.8** | **0.729** | 0.630 |
| TABLeT FT | **95.3** | 0.986 | **0.958** | **0.650** | **0.655** | **0.552** | **0.796** | **0.732** | **0.435** | 65.8 | 0.722 | **0.639** |

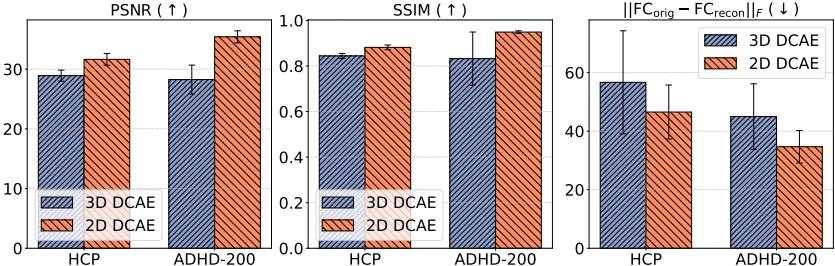

Figure 3: Reconstruction Quality of 3D DCAE and 2D DCAE.

## 4.4 COMPARISON OF 2D NATURAL IMAGE DCAE AND 3D FMRI-TRAINED DCAE

**Reconstruction Quality.** To evaluate the suitability of different tokenizers, we compared the reconstruction performance of 2D DCAE and 3D DCAE directly on fMRI data. Specifically, we computed PSNR and SSIM for each 3D volume and then averaged the results across all time steps and subjects. We also compared the difference in FC matrix between the original fMRI data ($FC_{orig}$) and its reconstruction ($FC_{recon}$): $||FC_{orig} - FC_{recon}||_F$. 3D DCAE was trained with the UKB dataset; a detailed training procedure is provided in Sec. B. To assess generalizability, we deliberately excluded HCP and ADHD-200 from the training set. The reconstructions from the three slicing axes were averaged for 2D DCAE.

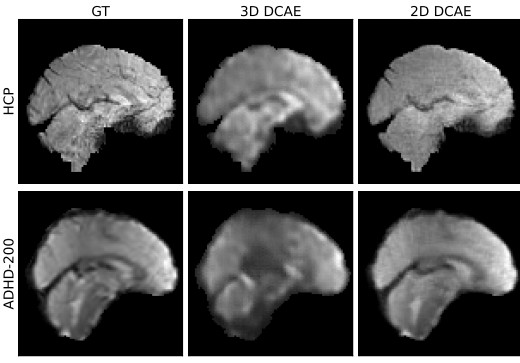

Figure 2: Visualization of reconstructions from 3D DCAE and 2D DCAE.

Remarkably, the 2D DCAE achieved higher reconstruction quality than the 3D DCAE trained directly on fMRI data. We believe that this finding suggests that the 2D DCAE preserves the information in fMRI data more effectively than the 3D DCAE without additional fine-tuning, indicating its potential as an effective tokenizer for fMRI data. As a side note, we also attempted to fine-tune the 2D DCAE with fMRI data while freezing different parts of the autoencoder, but discovered that any fine-tuning consistently harmed the reconstruction quality. We presume this is because our fMRI dataset is relatively small and homogeneous compared to the dataset the model is trained for, potentially harming generic filters crucial for the model's generalization capabilities.

**Training Performance.** We also compared models trained with latents from the 3D DCAE and the 2D DCAE. As shown in Tab. 1, both models achieve competitive performance, with the 2D DCAE outperforming the 3D counterpart in most cases.

Table 3: Performance comparison between TABLeT with latents from 3D DCAE and 2D DCAE on HCP and ADHD-200.

| Tokenizer | HCP | | | | | | | | | ADHD-200 | | |
|---|---|---|---|---|---|---|---|---|---|---|---|---|
| | Sex | | | Age | | | Intelligence | | | Diagnosis | | |
| | ACC | AUC | F1 | MSE | MAE | $\rho$ | MSE | MAE | $\rho$ | ACC | AUC | F1 |
| 3D DCAE | 92.2 | 0.973 | 0.929 | **0.767** | **0.693** | **0.475** | 0.869 | 0.755 | 0.387 | 65.8 | 0.711 | **0.644** |
| 2D DCAE | **93.8** | **0.987** | **0.943** | 0.773 | 0.705 | 0.473 | **0.835** | **0.741** | **0.392** | 65.8 | **0.729** | 0.630 |

## 4.5 MEMORY AND COMPUTATIONAL EFFICIENCY

As the development of TABLeT was motivated by the goal of making a fast and efficient voxel-based model, here we conduct a quantitative analysis to compare the memory and computational efficiency between TABLeT and SwiFT. To ensure a fair comparison, all tests were performed on a single GPU, and the batch size of both models was fixed to 4. We were only able to run SwiFT up to $T = 50$ due to memory limitations. At $T = 50$, compared to SwiFT, TABLeT is 7.33 times more memory efficient, and trains 3.8 times faster. With a similar memory budget ($\sim$30GB), $T$ can be extended nearly tenfold between SwiFT ($T = 40$) and TABLeT ($T = 384$).

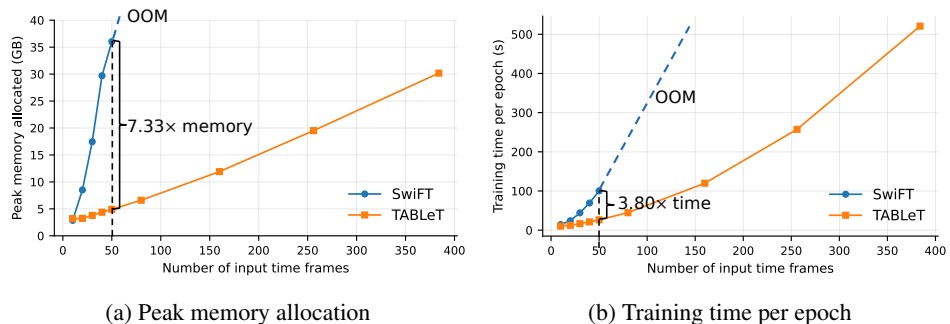

(a) Peak memory allocation  (b) Training time per epoch

Figure 4: Comparison of (a) memory and (b) training time, between TABLeT and SwiFT.

## 4.6 ADDITIONAL ABLATION STUDIES

**Effect of Aggregation of Three Axes.** We examined the effect of axis aggregation to better understand its effect: we compared models trained with fMRI tokens derived from a single axis alongside the model with aggregated tokens. As shown in Tab. 4, the performance of TABLeT varies depending on the chosen axis for single-axis models. In contrast, our aggregated version consistently achieves strong performance across tasks, eliminating the dependence on any particular slicing axis. These results represent why we chose to aggregate all three axes instead of using a single axis.

Table 4: Effect of the choice of slicing axis and aggregation of the three axes on classification and regression tasks. The best results are **bolded** and the second best results are underlined.

| Axis | UKB | | | | | |
|---|---|---|---|---|---|---|
| | Sex | | | Age | | |
| | ACC | AUC | F1 | MSE | MAE | $\rho$ |
| Sagittal | 97.3$_{\pm0.9}$ | 0.996$_{\pm0.001}$ | 0.971$_{\pm0.009}$ | 0.369$_{\pm0.015}$ | 0.486$_{\pm0.010}$ | 0.796$_{\pm0.008}$ |
| Coronal | 97.1$_{\pm0.4}$ | 0.996$_{\pm0.002}$ | 0.969$_{\pm0.004}$ | 0.435$_{\pm0.012}$ | 0.525$_{\pm0.008}$ | 0.756$_{\pm0.009}$ |
| Axial | 97.3$_{\pm0.4}$ | 0.997$_{\pm0.000}$ | 0.971$_{\pm0.004}$ | 0.410$_{\pm0.020}$ | 0.509$_{\pm0.014}$ | 0.771$_{\pm0.013}$ |
| All | **97.7**$_{\pm0.2}$ | **0.998**$_{\pm0.000}$ | **0.976**$_{\pm0.002}$ | **0.340**$_{\pm0.011}$ | **0.466**$_{\pm0.010}$ | **0.814**$_{\pm0.009}$ |

| Axis | HCP | | | | | |
|---|---|---|---|---|---|---|
| | Sex | | | Age | | |
| | ACC | AUC | F1 | MSE | MAE | $\rho$ |
| Sagittal | 91.3$_{\pm3.6}$ | 0.972$_{\pm0.017}$ | 0.920$_{\pm0.033}$ | 0.783$_{\pm0.111}$ | 0.721$_{\pm0.041}$ | 0.458$_{\pm0.076}$ |
| Coronal | 93.6$_{\pm1.7}$ | 0.981$_{\pm0.007}$ | 0.941$_{\pm0.015}$ | 0.855$_{\pm0.053}$ | 0.745$_{\pm0.023}$ | 0.376$_{\pm0.048}$ |
| Axial | 92.3$_{\pm3.0}$ | 0.979$_{\pm0.008}$ | 0.930$_{\pm0.028}$ | **0.748**$_{\pm0.056}$ | 0.711$_{\pm0.015}$ | 0.470$_{\pm0.040}$ |
| All | **93.8**$_{\pm0.9}$ | **0.987**$_{\pm0.003}$ | **0.943**$_{\pm0.008}$ | 0.773$_{\pm0.077}$ | **0.705**$_{\pm0.038}$ | **0.473**$_{\pm0.053}$ |

| Axis | HCP | | | ADHD-200 | | |
|---|---|---|---|---|---|---|
| | Intelligence | | | Diagnosis | | |
| | MSE | MAE | $\rho$ | ACC | AUC | F1 |
| Sagittal | 0.842$_{\pm0.058}$ | 0.744$_{\pm0.028}$ | **0.401**$_{\pm0.060}$ | **65.8**$_{\pm2.3}$ | 0.715$_{\pm0.026}$ | **0.633**$_{\pm0.032}$ |
| Coronal | 0.850$_{\pm0.057}$ | 0.749$_{\pm0.029}$ | 0.381$_{\pm0.065}$ | 63.5$_{\pm3.1}$ | 0.707$_{\pm0.036}$ | 0.621$_{\pm0.040}$ |
| Axial | 0.896$_{\pm0.070}$ | 0.773$_{\pm0.033}$ | 0.309$_{\pm0.072}$ | 64.3$_{\pm2.5}$ | 0.713$_{\pm0.022}$ | 0.622$_{\pm0.034}$ |
| All | **0.835**$_{\pm0.053}$ | **0.741**$_{\pm0.028}$ | 0.392$_{\pm0.062}$ | **65.8**$_{\pm3.5}$ | **0.728**$_{\pm0.020}$ | 0.630$_{\pm0.038}$ |

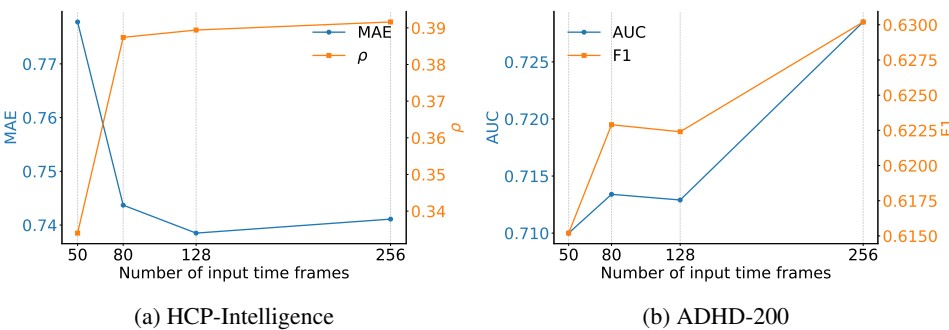

(a) HCP-Intelligence        (b) ADHD-200

Figure 5: Performance of TABLeT on HCP-Intelligence and ADHD-200 with varying $T$.

**Effect of $T$.** As shown in Tab. 1, modeling longer-range temporal dynamics can improve performance on the HCP-Intelligence and ADHD diagnosis tasks. To explore this further, we varied the $T$ of TABLeT and evaluated the corresponding performance. Interestingly, Fig. 5 reveals a clear positive trend between performance and $T$. We believe that investigating the relationship between $T$ and model performance across diverse tasks represents a promising direction for future research.

### 4.7 INTERPRETATION RESULTS

One advantage of voxel-based methods is that the models are interpretable, since the entire process from voxel to prediction is differentiable. To test the interpretability of TABLeT, we used Integrated Gradients (IG) (Sundararajan et al., 2017) for visualization of highly contributing areas for sex-classification. We used female test subjects in the HCP-Sex task who are correctly classified with TABLeT with high confidence ($\geq 75\%$), and computed the IG map of the first frame from each subject, then averaged it.

Fig. 6 shows that TABLeT mainly focuses on the medial prefrontal gyrus (mPFC), posterior cingulate cortex (PCC), precuneus (PCu), and thalamus (Thal.), where the regions are implicated in brain sex difference literature (Ficek-Tani et al., 2023; Ryali et al., 2024; Weis et al., 2020; Salinas et al., 2012).

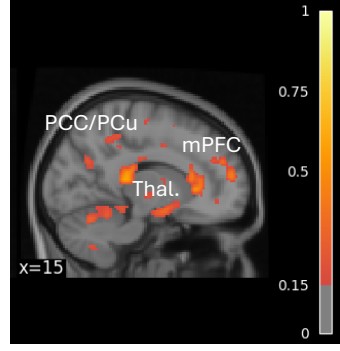

Figure 6: IG map of TABLeT.

## 5 CONCLUSION & LIMITATIONS

We presented TABLeT, a simple and efficient framework that leverages a 2D autoencoder trained on natural images to tokenize fMRI volumes. This tokenization enables long-range temporal modeling with Transformers while substantially reducing memory and computational costs. Experiments on UKB, HCP, and ADHD-200 demonstrated that TABLeT achieves competitive or superior performance compared to both ROI-based and voxel-based baselines. In addition, pretraining of TABLeT with masked token modeling further improved downstream task performance.

Despite these advantages, our study has several limitations. First, TABLeT tokenizes each frame of the fMRI time series independently. While effective, this process may disrupt subtle temporal dynamics. Future work could explore tokenization strategies that directly incorporate temporal dependencies, especially in tasks where fine-grained dynamics are critical. Second, TABLeT processes all tokens jointly, without explicit modeling of their spatial or temporal structure. Architectures designed to leverage spatial and temporal alignment between tokens may further enhance the ability to capture the spatiotemporal dynamics inherent in fMRI data.

Nevertheless, we believe our study suggests a promising approach, bridging natural image processing and medical imaging, and enabling scalable, efficient spatiotemporal modeling of brain activity.

## REPRODUCIBILITY STATEMENT

To ensure the reproducibility of our results, we have provided detailed descriptions and resources throughout the paper and the appendices. The 2D DCAE model utilized in our experiments is publicly available on Hugging Face under the identifier `mit-han-lab/dc-ae-f32c32-in-1.0`, as detailed in Sec. 3.1. The preprocessing pipeline for the fMRI dataset, clarifying the alignment to MNI space, is outlined in Sec. 4.1. Implementation details, such as GPU specifications and training configurations, are specified in Sec. A. Furthermore, the hyperparameters and training procedures of the 3D DCAE, which was built upon the 2D DCAE, are described in Sec. B.

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

# Efficient Modeling of Long-range fMRI Dynamics with a 2D Natural Image Autoencoder

## Appendix

### THE USE OF LARGE LANGUAGE MODELS (LLMS)

We utilized LLMs for the purpose of polishing our manuscript only.

## A  IMPLEMENTATION DETAILS

All experiments were conducted on the NVIDIA A100-40GB and RTX A6000 GPUs. We used `fp16` mixed precision for the training of all models except for TFF due to NaN error during training.

We used `BCEWithLogitsLoss` for the classification task, and used `pos-weight` option for the ADHD task to account for class imbalance. We used `L1Loss` for the regression tasks.

For the voxel-based models, TFF, SwiFT, and TABLeT, training was performed by randomly sampling consecutive 3D volumes. For evaluation, following Kim et al. (2023), we computed the final prediction by averaging the model outputs over all possible windows starting from the first frame.

**Shared Settings**  We used the following strategy for all of the experiments, unless explicitly stated.

- `Optimizer`: AdamW using a cosine decay learning rate scheduler, with weight decay of $10^{-2}$.
- `Hyperparameter Search`: For the UKB-Sex and HCP-Sex tasks, we searched the hyperparameter based on the validation AUROC for each model. For the UKB-Age, HCP-Age, and HCP-Intelligence tasks, we searched based on the validation MAE. For ADHD, we searched based on the validation loss to consider the `pos-weight` for the class imbalance.
- `Early Stopping`: We chose the early-stopped model for the BrainNetCNN, BNT, meanMLP, Brain-JEPA, and TFF by default. As we observed that SwiFT and TABLeT are more stable during training, we report results from the final epoch for all tasks.

**XGBoost**  We grid searched for hyperparameter tuning of XGBoost for the following.

- `Maximum depth`: Chosen between 3 and 5
- `Minimal child weight`: Chosen between 1 and 7
- `Gamma`: Chosen between 0.0 and 0.4
- `Learning rate`: Chosen between 0.05 and 0.3
- `Colsample by tree`: Chosen between 0.6 and 0.9

**BrainNetCNN**  We trained BrainNetCNN with the following setup:

- `Learning rate`: Chosen between $1 \times 10^{-6}$ and $2 \times 10^{-4}$
- `Batch size`: 64
- `Epochs`: 100 epochs of training

**Brain Network Transformer**  We trained Brain Network Transformer with the following setup:

- `Learning rate`: Chosen between $1 \times 10^{-6}$ and $2 \times 10^{-4}$
- `Batch size`: 64
- `Epochs`: 100 epochs of training

**meanMLP**  We trained meanMLP with the following setup:

- `Learning rate`: Chosen between $1 \times 10^{-4}$ and $1 \times 10^{-2}$
- `Batch size`: 32
- `Epochs`: 100 epochs of training

**Brain-JEPA** We trained Brain-JEPA from scratch for fair comparison with the following setup.

- `Learning rate:` Chose between $1 \times 10^{-5}$ and $7 \times 10^{-4}$.
- `Batch size:` 16
- `Epochs:` 50 epochs of training

**TFF** We trained TFF with the following setup:

- Phase 1
  - `Learning rate:` $3 \times 10^{-3}$ for UKB, ADHD, and $7 \times 10^{-4}$ for HCP
  - `Batch size:` 4
  - `Epochs:` 100 epochs of training
- Phase 2
  - `Learning rate:` $1 \times 10^{-5}$ for UKB, ADHD, and chosen between $1 \times 10^{-5}$ and $1 \times 10^{-6}$
  - `Batch size:` 2
  - `Epochs:` 50 epochs of training
- Fine-tuning
  - `Learning rate:` Chosen between $1 \times 10^{-5}$ and $1 \times 10^{-6}$ for UKB and ADHD, chosen between $3 \times 10^{-7}$ and $1 \times 10^{-6}$ for HCP,
  - `Batch size:` 4
  - `Epochs:` 10 epochs of training for UKB-Sex, 20 epochs of training for HCP, UKB-Age, and 30 epochs of training for ADHD.

**SwiFT** We trained SwiFT with the following setup:

- `Learning rate:` Chosen between $1 \times 10^{-6}$ and $5 \times 10^{-5}$
- `Batch size:` 4
- `Epochs:` 25 epochs of training for UKB, HCP, 30 epochs for ADHD.

**TABLeT** We trained TABLeT with the following setup:

- `Learning rate:` Chosen between $3 \times 10^{-7}$ and $5 \times 10^{-5}$
- `Batch size:` 4
- `Epochs:` 50 epochs of training for HCP-Sex, HCP-Intelligence, ADHD, 30 epochs for age regression, 15 epochs for UKB-Sex.

## B   Training Details of 3D fMRI-trained DCAE

We developed 3D DCAE by adapting the architecture of 2D DCAE (Chen et al., 2025) to handle 3D volume inputs. To achieve this, we replaced 2D convolutional layers with 3D convolutional layers and adjusted components such as RMS normalization, batch normalization, `PixelUnshuffle`, and `PixelShuffle` to process 3D data effectively. The model was configured with 1 input channel, 1024 latent channels, encoder-decoder width of `[16, 64, 256, 256, 1024, 1024]`, and encoder-decoder depth of `[0, 2, 2, 5, 5, 5]`, to make the same compression ratio as the 2D DCAE.

For training, we utilized a dataset of 8,178 subjects from the UK-Biobank, splitting it into training and validation sets with a 9:1 ratio and stratification based on sex and age. The model was trained for 100 epochs with an initial learning rate of $4 \times 10^{-5}$, which was gradually reduced using `ReduceLROnPlateau` scheduler. During each epoch, we randomly selected a single fMRI frame from the full set of frames for each subject to train the model. The training process used $\mathcal{L}_2$ reconstruction loss and the AdamW optimizer with a weight decay of $1 \times 10^{-4}$.

As the training curve in Fig. 7 shows, we made every effort to train the 3D DCAE model to achieve the best performance and ensure full convergence, for fair comparison.

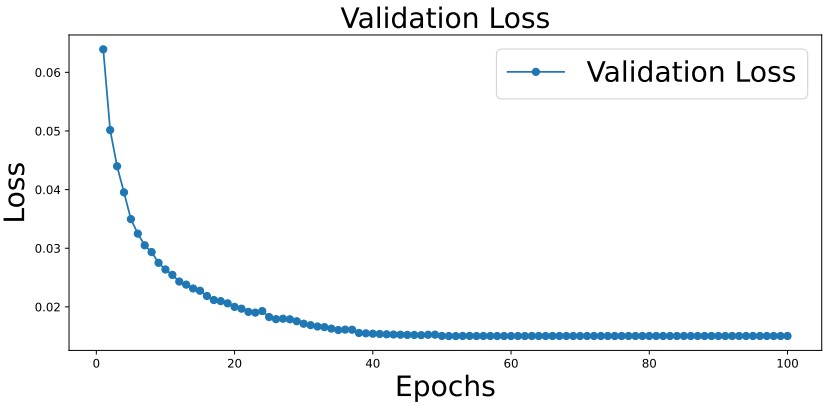

Figure 7: Validation Loss Curve for Training of 3D DCAE.

## C   DETAILED EXPERIMENTAL RESULTS

We provide the results reported in the manuscript with the standard deviation in Tab. 5, Tab. 6, and Tab. 7.

Table 5: Experimental results with standard deviation on UKB.

| Method | UKB | | | | | |
|---|---|---|---|---|---|---|
| | Sex | | | Age | | |
| | ACC | AUC | F1 | MSE | MAE | $\rho$ |
| XGBoost | $84.1_{\pm 1.7}$ | $0.916_{\pm 0.012}$ | $0.830_{\pm 0.019}$ | $0.698_{\pm 0.013}$ | $0.686_{\pm 0.008}$ | $0.553_{\pm 0.018}$ |
| BrainNetCNN | $91.7_{\pm 0.9}$ | $0.969_{\pm 0.007}$ | $0.912_{\pm 0.009}$ | $0.597_{\pm 0.017}$ | $0.618_{\pm 0.007}$ | $0.647_{\pm 0.012}$ |
| BNT | $92.4_{\pm 0.9}$ | $0.980_{\pm 0.003}$ | $0.919_{\pm 0.009}$ | $0.541_{\pm 0.016}$ | $0.588_{\pm 0.011}$ | $0.685_{\pm 0.011}$ |
| meanMLP | $87.7_{\pm 1.8}$ | $0.949_{\pm 0.009}$ | $0.869_{\pm 0.020}$ | $0.672_{\pm 0.031}$ | $0.662_{\pm 0.016}$ | $0.586_{\pm 0.027}$ |
| Brain-JEPA | $86.8_{\pm 0.6}$ | $0.943_{\pm 0.004}$ | $0.862_{\pm 0.007}$ | $0.688_{\pm 0.017}$ | $0.669_{\pm 0.008}$ | $0.574_{\pm 0.018}$ |
| TFF ($T = 20$) | $98.3_{\pm 0.4}$ | $0.998_{\pm 0.001}$ | $0.982_{\pm 0.004}$ | $0.440_{\pm 0.029}$ | $0.525_{\pm 0.015}$ | $0.760_{\pm 0.015}$ |
| SwiFT ($T = 20$) | $97.4_{\pm 0.3}$ | $0.998_{\pm 0.001}$ | $0.972_{\pm 0.003}$ | $0.366_{\pm 0.005}$ | $0.480_{\pm 0.007}$ | $0.800_{\pm 0.004}$ |
| SwiFT ($T = 50$) | $98.1_{\pm 0.4}$ | $0.999_{\pm 0.001}$ | $0.980_{\pm 0.005}$ | $0.364_{\pm 0.004}$ | $0.477_{\pm 0.005}$ | $0.802_{\pm 0.003}$ |
| TABLeT ($T = 256$) | $97.6_{\pm 0.2}$ | $0.998_{\pm 0.000}$ | $0.975_{\pm 0.002}$ | $0.340_{\pm 0.011}$ | $0.466_{\pm 0.010}$ | $0.814_{\pm 0.009}$ |

Table 6: Experimental results with standard deviation on HCP sex classification and age regression.

| Method | HCP | | | | | |
|---|---|---|---|---|---|---|
| | Sex | | | Age | | |
| | ACC | AUC | F1 | MSE | MAE | $\rho$ |
| XGBoost | $82.2_{\pm 2.5}$ | $0.890_{\pm 0.028}$ | $0.837_{\pm 0.025}$ | $0.859_{\pm 0.074}$ | $0.769_{\pm 0.033}$ | $0.296_{\pm 0.112}$ |
| BrainNetCNN | $86.3_{\pm 4.9}$ | $0.937_{\pm 0.027}$ | $0.866_{\pm 0.049}$ | $0.847_{\pm 0.097}$ | $0.749_{\pm 0.040}$ | $0.372_{\pm 0.097}$ |
| BNT | $86.3_{\pm 3.0}$ | $0.935_{\pm 0.026}$ | $0.872_{\pm 0.030}$ | $0.794_{\pm 0.051}$ | $0.719_{\pm 0.027}$ | $0.444_{\pm 0.055}$ |
| meanMLP | $84.5_{\pm 2.5}$ | $0.915_{\pm 0.018}$ | $0.855_{\pm 0.028}$ | $0.846_{\pm 0.056}$ | $0.751_{\pm 0.030}$ | $0.370_{\pm 0.087}$ |
| Brain-JEPA | $73.9_{\pm 3.2}$ | $0.809_{\pm 0.018}$ | $0.761_{\pm 0.043}$ | $0.814_{\pm 0.037}$ | $0.746_{\pm 0.009}$ | $0.369_{\pm 0.046}$ |
| TFF ($T = 20$) | $88.1_{\pm 5.0}$ | $0.937_{\pm 0.055}$ | $0.892_{\pm 0.042}$ | $0.888_{\pm 0.062}$ | $0.779_{\pm 0.036}$ | $0.246_{\pm 0.061}$ |
| SwiFT ($T = 20$) | $93.1_{\pm 0.5}$ | $0.978_{\pm 0.008}$ | $0.937_{\pm 0.004}$ | $0.776_{\pm 0.043}$ | $0.719_{\pm 0.015}$ | $0.450_{\pm 0.031}$ |
| SwiFT ($T = 50$) | $92.2_{\pm 1.1}$ | $0.972_{\pm 0.014}$ | $0.929_{\pm 0.010}$ | $0.764_{\pm 0.092}$ | $0.699_{\pm 0.047}$ | $0.460_{\pm 0.071}$ |
| TABLeT ($T = 256$) | $93.8_{\pm 0.9}$ | $0.987_{\pm 0.003}$ | $0.943_{\pm 0.008}$ | $0.773_{\pm 0.077}$ | $0.705_{\pm 0.038}$ | $0.473_{\pm 0.053}$ |
| TABLeT (3D DCAE) | $92.2_{\pm 1.7}$ | $0.973_{\pm 0.010}$ | $0.929_{\pm 0.014}$ | $0.767_{\pm 0.118}$ | $0.693_{\pm 0.043}$ | $0.475_{\pm 0.076}$ |
| TABLeT (FT) | $95.3_{\pm 1.3}$ | $0.986_{\pm 0.005}$ | $0.958_{\pm 0.011}$ | $0.650_{\pm 0.045}$ | $0.655_{\pm 0.024}$ | $0.552_{\pm 0.032}$ |

## D   DETAILED DATA DESCRIPTION

We provide a detailed description of each dataset used in our study in Tab. 8.

Table 7: Main experimental results with standard deviation on HCP intelligence regression and ADHD diagnosis.

| Method | HCP Intelligence | | | ADHD-200 Diagnosis | | |
|---|---|---|---|---|---|---|
| | MSE | MAE | $\rho$ | ACC | AUC | F1 |
| XGBoost | $0.908_{\pm 0.054}$ | $0.779_{\pm 0.023}$ | $0.292_{\pm 0.099}$ | $62.3_{\pm 2.5}$ | $0.650_{\pm 0.036}$ | $0.555_{\pm 0.031}$ |
| BrainNetCNN | $0.967_{\pm 0.119}$ | $0.788_{\pm 0.044}$ | $0.286_{\pm 0.112}$ | $59.2_{\pm 10.7}$ | $0.640_{\pm 0.095}$ | $0.545_{\pm 0.118}$ |
| BNT | $0.920_{\pm 0.092}$ | $0.778_{\pm 0.054}$ | $0.318_{\pm 0.083}$ | $63.6_{\pm 5.4}$ | $0.677_{\pm 0.062}$ | $0.624_{\pm 0.057}$ |
| meanMLP | $0.887_{\pm 0.076}$ | $0.767_{\pm 0.028}$ | $0.340_{\pm 0.045}$ | $56.8_{\pm 6.8}$ | $0.617_{\pm 0.067}$ | $0.532_{\pm 0.095}$ |
| Brain-JEPA | $0.959_{\pm 0.091}$ | $0.799_{\pm 0.033}$ | $0.171_{\pm 0.051}$ | – | – | – |
| TFF ($T = 20$) | $0.898_{\pm 0.022}$ | $0.767_{\pm 0.018}$ | $0.312_{\pm 0.088}$ | $63.3_{\pm 2.3}$ | $0.700_{\pm 0.028}$ | $0.608_{\pm 0.030}$ |
| SwiFT ($T = 20$) | $0.940_{\pm 0.111}$ | $0.782_{\pm 0.044}$ | $0.297_{\pm 0.080}$ | $63.3_{\pm 3.7}$ | $0.693_{\pm 0.030}$ | $0.623_{\pm 0.033}$ |
| SwiFT ($T = 50$) | $0.865_{\pm 0.093}$ | $0.758_{\pm 0.046}$ | $0.354_{\pm 0.070}$ | $63.9_{\pm 3.2}$ | $0.701_{\pm 0.032}$ | $0.627_{\pm 0.030}$ |
| TABLeT ($T = 256$) | $0.835_{\pm 0.053}$ | $0.741_{\pm 0.028}$ | $0.392_{\pm 0.062}$ | $65.8_{\pm 3.5}$ | $0.729_{\pm 0.029}$ | $0.630_{\pm 0.038}$ |
| TABLeT (3D DCAE) | $0.869_{\pm 0.077}$ | $0.755_{\pm 0.032}$ | $0.387_{\pm 0.078}$ | $65.8_{\pm 1.7}$ | $0.711_{\pm 0.026}$ | $0.644_{\pm 0.022}$ |
| TABLeT (FT) | $0.796_{\pm 0.051}$ | $0.732_{\pm 0.028}$ | $0.435_{\pm 0.046}$ | $65.8_{\pm 2.1}$ | $0.722_{\pm 0.022}$ | $0.639_{\pm 0.031}$ |

Table 8: Demographic information of the datasets used in our study

| Category | UKB | HCP | ADHD-200 |
|---|---|---|---|
| Number of subjects | 8,178 | 1,061 | 533 |
| Sex | | | |
|    Male, n (%) | 4,295 (52.5%) | 488 (46.0%) | 207 (38.8%) |
|    Female, n (%) | 3,883 (47.5%) | 573 (54.0%) | 325 (61.0%) |
|    N/A, n (%) | – | – | 1 (0.2%) |
| Age (years) | $54.98_{\pm 7.53}$ | $28.79_{\pm 3.70}$ | $11.94_{\pm 3.40}$ |
| Intelligence | – | $113.32_{\pm 20.50}$ | – |
| Diagnosed, n (%) | – | – | 236 (44.3%) |

