# OpenReview forum: "Efficient Modeling of Long-range fMRI Dynamics with a 2D Natural Image Autoencoder"
_ICLR.cc/2026/Conference — ICLR 2026 Conference Withdrawn Submission_

### Official Review · Reviewer_KbRQ · 2025-10-29

**Soundness:** 3
**Presentation:** 3
**Contribution:** 2
**Rating:** 4
**Confidence:** 4

**Summary:**

The paper Efficient Modeling of Long-Range fMRI Dynamics with a 2D Natural Image Autoencoder presents TABLeT, a method for large-scale spatiotemporal fMRI modeling that replaces voxel-level processing with tokenized representations derived from a pretrained 2D natural image autoencoder. Each 3D fMRI frame is compressed into a compact latent representation, which allows a Transformer encoder to process long temporal sequences efficiently. The paper reports improved performance and significantly lower memory use compared to voxel-based baselines such as SwiFT and TFF across multiple datasets (UK Biobank, HCP, ADHD-200). It also demonstrates a self-supervised masked token pretraining scheme that enhances downstream prediction tasks. The approach highlights a novel cross-domain idea—reusing natural image autoencoders for brain imaging—but its conceptual contribution remains incremental, as it mainly focuses on computational efficiency rather than advancing neuroscientific modeling or representation theory.

**Strengths:**

1. Innovative cross-domain use of pretrained 2D image autoencoders for fMRI compression.

2. Substantial improvements in training efficiency and scalability to long sequences.

3. Strong empirical validation across multiple datasets and tasks.

4. Inclusion of self-supervised pretraining and ablations adds robustness to the study.

5. Clear visualizations and interpretability demonstrations using integrated gradients.

**Weaknesses:**

1. Novelty is mainly technical; the approach does not significantly enhance neuroscientific insight or modeling depth.

2. Using 2D autoencoders trained on natural images introduces domain mismatch, risking information distortion.

3. The biological and functional interpretability claims are underdeveloped and not rigorously supported.

4. The pretraining improvement is marginal, and theoretical justification for why the natural-image prior works is weak.

5. Long-term implications for real clinical or neurocognitive modeling are not explored.

6. Lacks comparison with recent neuroimaging-specific foundation models such as BrainLM or Brain-JEPA.

**Questions:**

1. Why would a 2D natural image autoencoder preserve brain-specific structures better than a 3D fMRI-trained model?

2. Could domain adaptation or finetuning improve results without hurting reconstruction quality?

3. How generalizable is TABLeT to other neuroimaging modalities or non-resting-state data?

4. Are the compressed representations neurobiologically interpretable, or purely statistical tokens?

5. How does the model behave under varying spatial resolutions or scanner noise conditions?

6. How do you ensure that masked token pretraining improves rather than biases downstream learning?

7. What are the implications of using natural image features for medical interpretability and ethics?

---

### Official Review · Reviewer_D4kX · 2025-10-29

**Soundness:** 4
**Presentation:** 3
**Contribution:** 3
**Rating:** 8
**Confidence:** 4

**Summary:**

The paper introduces a model for predicting participants attributes based on RS-fMRI measurements.
The proposed  model uses a pretrained image autoencoder for extracting embeddings from fMRI frames.
This embeddings are aggregating into tokens which are passed into a transformer model.
The transformer model is pre-trained in a self-supervised manner by token masking.
Finally the model is trained and evaluated on predicting attributes like sex, age, IQ and ADHD diagnosis.

**Strengths:**

- The method is competitive and outperforms on a wide range of metrics.
- The idea of treating the fMRI slice as image is neat, and works well.
- The method is resource efficient.

**Weaknesses:**

The improvements over SWIFT are not big.

**Questions:**

1) Are 256 frames necessary, would taking a number of shorter windows and averaging over the output yield similar performance?
2) I think an ablation figure on window length would be beneficial. Specifically, it would strengthen the paper by indicating why resource efficiency is important.
3) more of suggestion, fine-tuning the DCAE might give a performance boost.

---

### Official Review · Reviewer_Liny · 2025-10-30

**Soundness:** 2
**Presentation:** 3
**Contribution:** 2
**Rating:** 2
**Confidence:** 4

**Summary:**

This paper proposes TABLeT, a Transformer-based model designed for fMRI time-series analysis. The core workflow of the model involves two main stages. First, a 2D Deep Convolutional Autoencoder (DCAE) is used to perform spatial compression on each fMRI frame, transforming it into a low-dimensional feature vector. Second, these sequences of feature vectors are pre-trained on a large-scale dataset (UK Biobank) using a self-supervised task of masked signal reconstruction. Finally, the pre-trained model is fine-tuned and evaluated on several downstream tasks, such as sex classification and age regression, where it is reported to achieve competitive results.

**Strengths:**

1. **Significance of the Problem**: The paper addresses the important goal of building a foundation model for fMRI analysis, a research direction with significant scientific value and immense potential in neuroscience and clinical research. Exploring foundation models suitable for large-scale brain functional data is a frontier topic in the field.

2. **Comprehensive Experimental Evaluation**: The authors conduct pre-training on a large-scale, high-quality dataset (UK Biobank) and perform extensive evaluations on multiple downstream tasks of varying natures. The experimental setup is comprehensive and provides good support for assessing the model's generalization capabilities.

3. **Clarity and Presentation**: The paper is well-organized and clearly written, making it easy for readers to understand the proposed methodology, experimental procedures, and core arguments.

**Weaknesses:**

1. **Limited Methodological Novelty**: The core methodology of this paper is primarily a direct application of established paradigms, lacking substantial innovation tailored to the unique spatio-temporal characteristics of fMRI data. The masked signal reconstruction strategy is conceptually derived from BERT in NLP and MAE in computer vision. The authors apply it to fMRI time-series without proposing specific improvements to handle the complex spatio-temporal dependencies in brain signals. Similarly, using an autoencoder for spatial compression to fit the input requirements of a Transformer is a common design pattern. For a top-tier conference like ICLR, a more original methodological contribution is expected, rather than an incremental combination of existing technologies.

2. **Insufficient Baseline Comparisons**: The experimental section fails to provide a sufficient comparison against several recent, high-performing baseline models in the field. For instance, recent works such as [1][2][3][4] offer new perspectives and strong performance benchmarks for fMRI analysis and should have been included for comparison. Furthermore, classic yet robust machine learning methods, such as the Support Vector Machine (SVM), which are commonly used and effective in fMRI classification studies, are lacking.

3. **The Claim of "Efficiency" is Misleading and Poorly Supported**: The authors claim their model is "efficient," with the main argument being that the DCAE's spatial compression allows for processing longer time series. I argue this point is fundamentally flawed. This is not a real improvement in computational efficiency but rather a trade-off between spatial information fidelity and temporal sequence length. This compression inevitably leads to a loss of spatial information. This point is strongly supported by the results in Figure 5. When TABLeT and the baseline SwiFT use the same number of input frames (50), TABLeT's performance is worse than SwiFT's. This suggests that the performance advantage of TABLeT stems from processing more compressed temporal frames, not from its architecture being inherently superior or more efficient on a per-frame basis.

4. **The Motivation for the 2D DCAE Module is Unclear**: The paper's explanation and justification for choosing an autoencoder designed for 2D natural images to process 3D volumetric fMRI data are unconvincing. The paper fails to provide sufficient insight into why a 2D autoencoder would be effective for fMRI data, which possesses a complex 3D spatial structure. In Section 4.4, the authors justify their choice by comparing the reconstruction quality of a 2D DCAE versus a 3D DCAE. This is inappropriate, as it conflates perceptual-level reconstruction quality with semantic-level task performance. High-fidelity reconstruction does not necessarily translate to better performance on downstream tasks, as some high-frequency local details may be redundant or even act as noise for semantic tasks like classification or regression. As shown in Table 2, the performance difference between models using a 2D DCAE and a 3D DCAE on downstream tasks is marginal, which actually weakens the authors' argument for choosing the 2D version. More importantly, did the authors attempt to activate the 3D DCAE for direct training in downstream tasks? It's even possible to use the same 3D DCAE, simultaneously connecting it to two Transformer Encoders for classification and regression to train both tasks concurrently, preserving more effective information in the compressed tokens (age and gender may have independent effects on brain activity). If this approach proves more effective, it would directly overturn the motivation of choosing the compression module based on reconstruction results.

References:

[1] Hierarchical Spatio-Temporal State-Space Modeling for fMRI Analysis, In RECOMB 2025.

[2] CBrain: Cross-Modal Learning for Brain Vigilance Detection in Resting-State fMRI, In MICCAI 2025.

[3] A simple but tough-to-beat baseline for fMRI time-series classification, NeuroImage 303 (2024).

[4] BrainMT: A Hybrid Mamba-Transformer Architecture for Modeling Long-Range Dependencies in Functional MRI Data, In MICCAI 2025.

**Questions:**

The suggestion of improvement is stated in the Weaknesses. No further suggestions.

---

### Official Review · Reviewer_JqXe · 2025-10-30

**Soundness:** 3
**Presentation:** 3
**Contribution:** 1
**Rating:** 4
**Confidence:** 2

**Summary:**

This paper proposed a two-dimensional autoencode-based transformer to process the fMRI data into tokens. By incorporating the positional encoder, it aims to preserve information over the long-range spatialtemporal dynamics. The proposed methods have been applied on several datasets.

**Strengths:**

1. The problem considered in the paper is interesting.
2. The incorporation of positional encoding, and the idea of tokenizing the fMRI data is inspiring and interesting.
3. The paper is well-written and organized.

**Weaknesses:**

My primary concern is the novelty of the paper. From my current understanding, only the application of the positional encoding is new. Other parts, including 2-d auto-encoder and masked token modeling, are not newly proposed. Besides, the experimental result is not very convincing. First, it does not report the standard deviation. Besides, the improvement is not significant.

**Questions:**

In addition to fMRI, MEG and EEG are also of interest, as they typically exhibit lower temporal noise. Would it be possible to apply this method to MEG or EEG data as well?

---

### Note · Authors · 2025-11-13

I have read and agree with the venue's withdrawal policy on behalf of myself and my co-authors.